# Legal Mobilization and the Internationalization of Anticorruption Enforcement

**Mikkel Jarle Christensen**

Faculty of Law, University of Copenhagen, 2300 Copenhagen, Denmark; mjc@jur.ku.dk

**Abstract:** This article contributes a critical study of efforts to internationalize the investigation and prosecution of corruption. The efforts to internationalize anticorruption enforcement are visible, for instance, in calls for an International Anticorruption Court (IACC) or an Anticorruption Protocol to the United Nations Convention against Corruption (APUNCAC). Inspired by a historical sociological perspective, this article investigates mobilizations around these initiatives, how mobilizers frame their engagement, and the ideological context in which they operate. In particular, the article zooms in on elites and how they push for states to internationalize the investigation and prosecution of corruption. This article situates the efforts of these elites in a larger historical context and compares the push to internationalize anticorruption enforcement to earlier legal mobilizations in the field of international criminal justice focused on atrocity crimes.

**Keywords:** legal mobilization; corruption; sociology of law

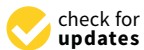

## 1. Introduction

Corruption affects societies across the world. These effects are demonstrated, for instance, by Transparency International's (TI) Corruption Perceptions Index (CPI). The annual CPI suggests that corruption is endemic in many societies: most pronounced in mid- and low-income countries, often the countries referred to as the global south, but also present in high and high–middle income countries in the global north. At the same time, the focus on perceptions of corruption is indicative of the difficulties of directly measuring corrupt practices and their effects.

In addition to affecting societies across the world, corruption often affects the state, both in terms of its perceived legitimacy and its functionality. In addition to its effects on the state and its institutions, corruption is also likely to impact the relation between states, and thus have effects at the international and even global level (Elliott 1997; Rotberg 2009). Whereas such effects can be difficult to disentangle empirically, corruption can help produce and reproduce weak states that are prone to exploitation linked to different forms of criminal activity (Kodila-Tedika and Bolito-Losembe 2014). For instance, the multi-billion dollar business of illegal drug trafficking is likely to both produce and profit from corrupt practices. Drug flows into high-income countries, including North America and Europe, often transit through states with weak institutions and high levels of corruption, for instance in Latin America (Bagley 1988, 1991; Tate 2015; White 2019), the Caribbean (Bowling 2010) and West African states such as Guinea-Bissau (Shaw 2015; Vigh 2017). War and atrocities committed during conflicts are also linked to corrupt practices, for instance with regard to extractive industries in countries characterized by violent and weaponized conflict, a dynamic conceptualized (and popularized) through the notion of "blood diamonds" (Bieri 2016; Marchuk 2009).

Over the past 30 years, a portion of the international community has mobilized to fight corruption and the problems associated with it. However, whereas international legal frameworks were developed to combat corruption, most notably the 2003 United Nations Convention against Corruption (UNCAC), these initiatives have, in general, relied upon self-enforcement by states parties. In response, a range of agents mobilized to

internationalize anticorruption enforcement. These agents pointed to the internationalization of enforcement as necessary for dealing with the perceived transnational effects of corruption and potential impact on international relations, peace and security. Supporters of internationalization assert that domestic enforcement efforts have, for various reasons, been inadequate.

The move to internationalize the fight against corruption is inspired by, and to some extent modeled on, prior international efforts to investigate and prosecute atrocity crimes (Drumbl 2007; Osiel 2009) also known as core international crimes (Clapham and Gaeta 2014). The perceived success of these earlier international initiatives supports a belief that the internationalization of anticorruption enforcement might be equally successful. Several of the agents involved in earlier initiatives were also active in the more recent effort to internationalize the fight against corruption by pushing for new enforcement mechanisms outside of the state.

However, the push to internationalize the investigation and prosecution of corruption occurs in a context characterized by a very different legal and political ideology when compared to the push to internationalize the enforcement of atrocity crimes that accelerated in the 1990s. There is a need to explore these differences and consider the possible implications. Ultimately, the question is whether the current context is adequate to foster and support the strong internationalization of anticorruption enforcement. If the context is adequate, then the precedents established regarding atrocity crimes are presumably relevant. However, if the context is inadequate, then those precedents are less meaningful.

In this article, I analyze efforts to mobilize international enforcement against core international crimes, then compare these with efforts to mobilize international investigation and prosecution of corruption. I describe the context of these efforts and the evolution of the literature regarding international enforcement. I then analyze international enforcement efforts against core crimes and mobilizations against corruption, focusing on the agents, patterns of mobilization and legal framing. The conclusion outlines the main differences and similarities between the two patterns of mobilization and points to the broader implications of the analysis.

## 2. Context

While much has been written on the topic of corruption, the focus of the current article is to contribute an analysis that compares the efforts to internationalize the investigation and prosecution of core international crimes, the context of their occurrence and the context how agents mobilize to internationalize anticorruption enforcement. This approach involves a historical, sociological analysis and comparison of the two contexts in which two, in some ways related, forms of legal mobilization were developed. I begin with a review of the previous literature focusing on efforts to mobilize the international community regarding anticorruption.

Previous studies highlighted mobilization around anti-corruption norms driven by NGOs such as TI (Gutterman 2014; Larmour 2005; Wang and Rosenau 2001). In response to a 'corruption eruption' (Naim 1997), institutional and civil society 'integrity warriors' (Sampson 2005; de Sousa et al. 2009) mobilized to raise awareness of the detrimental effects of corruption, advance transnational anticorruption norms, and develop novel legal frameworks and anticorruption enforcement strategies.

In this period, new legal frameworks were negotiated, including the UNCAC. The effectiveness of such frameworks, however, is debatable (Webb 2005; Weilert 2016). This period also saw the creation of other legal institutions and frameworks such as the Financial Action Task Force (FATF) (Alexander 2001; Nance 2018), which specifically aims to control money laundering, often linked to large-scale and systematic corruption, and the 1997 OECD Convention on Combating Bribery of Foreign Public Officials in International Business Transactions (D'Souza 2012). At a regional level, the Inter-American Convention against Corruption was finalized in 1996 (Altamirano 2006; Manfroni et al. 2003). The African Union adopted the Convention on Preventing and Combating Corruption in 2003

(Carr 2007; Olaniyan 2004). In the European Union (EU), different institutions work to combat corruption. The European Public Prosecutor's Office (EPPO) began operations in June 2021 (De Angelis 2019; Ligeti and Simonato 2013). Established after decades of scholarly and political debate (Delmas-Marty and De Angelis 1995), the EPPO has jurisdiction in 22 EU member states with regard to the misappropriation of EU funds. In addition to such initiatives, national anti-corruption units and agencies were created and political leaders focused their rhetoric on corruption. However, amid reports highlighting the inadequacy of established legal frameworks, activists increasingly proposed the internationalization of efforts to investigate and prosecute corruption.

The debate regarding the internationalization of criminal enforcement stretches back to the interwar era (Lewis 2014) when discussion arose regarding an international criminal court to judge the German Kaiser for waging an aggressive war (Schabas 2018). In 1937, an international criminal court was established, if only on paper, to deal with terrorism (Marston 2003). The debate continued throughout the Cold War period, primarily in the form of scholarly writing and rarely at the political level (Bassiouni 1987; Stone and Woetzel 1970). When political debates about an international criminal court were reanimated in the 1980s and 1990s, discussion focused on the types of crimes over which a court would have jurisdiction. These debates built on previous proposals for an international criminal court, but pushed for a different subject-matter jurisdiction. In the early 1980s, African states promoted the establishment of an international tribunal to adjudicate crimes associated with apartheid (Bassiouni and Derby 1981). Six years later, the Soviet Union again suggested an international criminal court to deal with terrorism (Gorbachev 1987). A 1989 proposal by Trinidad and Tobago pushed for drug trafficking to be at the core of the subject-matter jurisdiction of an international criminal court (Boister 1998).

The 1998 Rome Statute that created the International Criminal Court (ICC) focused on war crimes, crimes against humanity, genocide, and, after the Kampala Amendment to the Statute (Trahan 2011; Wenaweser 2010), the crime of aggression. The debate about which crimes international criminal courts ought to cover continued after the establishment of the permanent court (Boister 2012). Terrorism was once more proposed as a crime that could be included under the jurisdiction of the ICC or could fall under the jurisdiction of a new international criminal court (Arnold 2004; Cohen 2011), although such proposals seem to have stalled during recent years, perhaps due to the limited success of the Special Tribunal for Lebanon (STL) that deals with terrorism (Alamuddin et al. 2014; Burgis-Kasthala and Saouli 2021; Hillebrecht 2020). Initiatives to include additional crimes under the jurisdiction of the ICC also include ecocide (Crook and Short 2021; Gauger et al. 2012; Gray 1995; Hellman 2014; Higgins 2012), human trafficking (Moran 2014), piracy (Dutton 2010; Kazemi and Heidari 2019; O'Brien 2014), and, most relevant in the present context, corruption (Christensen 2017a). Legal scholars suggested that corruption could be prosecuted under a broad interpretation of the ICC's mandate, possibly as a crime against humanity (Bantekas 2006; Roht-Arriaza and Martinez 2019). This would significantly expand the jurisdiction of the ICC.

The present article builds broadly on research regarding legal mobilizations (Kessler 1990; McAdam 2010; McCann 1994; Scheingold 2010; Zemans 1983). Generally, this research takes a bottom-up perspective on the multitude of mobilizations that seek to place specific problems, ideas, and norms on the legal and political agenda. Legal mobilization efforts focus on the development of anticorruption norms and strategies that activists may use to gain traction on the political agenda. Mobilizations may involve litigation and were linked to cause lawyering (Sarat 1998). In other instances, anticorruption scholars focused on strategies employed by civil society organizations, including NGOs, trade unions and legal professional associations, to promote specific legal solutions (Burstein 1991; McCann 2008; Vanhala 2010). Inspired by network theory (Keck and Sikkink 1999; Sikkink 2011), frameworks related to legal mobilization and to broader social movement studies were used to analyze the development of international criminal law, including the negotiation of the Rome Statute (Glasius 2006).

This research is linked to wider theories of legal mobilization, its strategies, dynamics, and effects (Lehoucq and Taylor 2020; McCann 2017). Lehoucq and Taylor contribute a concise definition of legal mobilization and also provide a typology of legal mobilization that distinguishes between "*legal mobilization* as the use of the law in an explicit, self-conscious way through the invocation of a formal institutional mechanism, *legal framing* as the use of law in an explicit, self-conscious way to give meaning to an event, and *legal consciousness* as the implicit, nonarticulated use of law to give meaning to an event" (Lehoucq and Taylor 2020). To analyze the push for new internationalized approaches to anticorruption enforcement, I build on Lehoucq and Taylor's typology. In the present article, mobilizations are studied primarily through the (often elite) agents that invest in specific initiatives to internationalize, whereas frames are studied through the conceptual frameworks developed and deployed by these agents. In studying how specific frameworks are used in legal mobilizations, I build on the research regarding the ways that such schemes influence social movements (Benford and Snow 2000; Goffman 1974; Snow and Benford 2005). This research shows how frames are interactive and the subject of collective work of discursive construction and definition (Benford and Snow 2000). At least three levels of framing can be identified (Snow and Benford 1988): *diagnostic*, *prognostic,* and *motivational*. The background for such framings, conceptualized as legal consciousness by Lehoucq and Taylor, is studied through the reception of the norms and ideas generated by mobilizations around internationalization. Legal consciousness is difficult to identify empirically, compared to the agents active in mobilizations. However, it can be identified in the wider historical and societal context in which frames are developed, frames that may be in line with dominant ideologies and patterns of thinking or, at times, may explicitly or implicitly challenge such orthodoxies.

The study of legal mobilizations also touches on structural dynamics that shape the organization of new forms of enforcement and their influence on political agendas. Structural dynamics are visible, for instance, in the backgrounds of the agents driving specific enforcement strategies, how they are organized and the networks that they mobilize and influence. Their professional power, understood here as their ability to influence the political agenda, is structured partly by their own backgrounds and resources, and partly by the context in which they champion specific legal solutions. The focus on professional power is inspired by Pierre Bourdieu's sociology, in particular his concept of habitus (Bourdieu and Wacquant 1992). The concept of habitus captures the various forms of experience and expertise (or capital, to use Bourdieu's concept) that are embodied in agents, as well as how these embodied properties influence their views of the world. Habitus is a consequence of the upbringing, schooling and, typically, the professional experience of specific agents. It also has structuring effects on the normative perspective of those agents regarding specific topics and issues, something Bourdieu famously showed in relation to the distinctions between different French social groups and their cultural tastes (Bourdieu 2013). In the present context, the habitus of specific elite agents, in particular their professional expertise and resources, is likely to influence the ways that they mobilize around specific strategies to internationalize the enforcement of laws against specific crimes.

Inspired by the literature on legal mobilizations and ways they are linked to specific agents and social groups, I employ historical sociological case studies to analyze the effort to internationalize anticorruption enforcement. The focal point of this analysis is the agents who are active in these legal mobilizations, the historical context in which they operate, and the frames and strategies they deploy to make a case for specific solutions. The analysis debunks certain myths regarding the ways that activists influence political decision making regarding international enforcement efforts.

In the following sections, I contrast two sets of mobilizations. Since the internationalization of enforcement against atrocity crimes occurred prior to the push to internationalize anticorruption enforcement, I begin by analyzing the legal mobilizations that occurred around international core crimes, including how they were framed and connected to wider

patterns of thought. I contrast this with mobilizations and frames around internationalization of anticorruption enforcement. The analysis seeks to identify the roots of the current push to internationalize anticorruption enforcement and draw distinctions, in comparison with legal mobilizations around international criminal law focused on atrocity crimes.

### 3. Mobilizations behind the Internationalization of Enforcement against Core Crimes

Previous scholars highlighted the important role of legal mobilizations against international core crimes. Kathryn Sikkink demonstrated how NGOs in different parts of the world pushed states to reckon with past crimes (Sikkink 2011), often using criminal law technologies as part of a wider transnational surge of human rights prosecutions (Engle 2014). The use of criminal law as a governance tool, both within states and at the international level, gained political salience around the end of the Cold War. Discussion continued regarding various proposals for an international criminal court focused on apartheid, terrorism or drug trafficking. Before the 1998 Rome Statute was adopted, however, other criminal tribunals were created with narrow jurisdiction over atrocity crimes. The two UN ad hoc tribunals, the International Criminal Tribunal for the former Yugoslavia (ICTY) and the International Criminal Tribunal for Rwanda (ICTR), built momentum towards the use of criminal law as an international governance technology. The tribunals focused on what became known as core international crimes.

The creation of the ad hoc tribunals served, at least in part, to substitute for military intervention. The US, in particular, was reticent to send soldiers to the former Yugoslavia (Hazan 2004) and Rwanda, the latter following the US intervention in Somalia (Dotson 2016) that led to 18 American casualties and 73 wounded. However, as evidenced in African, Soviet and Caribbean proposals to create an international criminal court, the perception that criminal law could and should be used as an international governance tool was the object of broad mobilizations. This was also the case in Europe, and in leading European countries such as Germany, where criminal law was promoted to reckon with core crimes. Criminal law was also discussed as a governance mechanism for the European Communities (Delmas-Marty and De Angelis 1995; Delmas-Marty and Vervaele 2000). This idea was carried forward with the emergence of the enlarged EU. Both individual states and regional powers invested in criminal law as a governance mechanism in the 1990s, which helped pave the way for the ICTY and ICTR (Christensen 2017b). In contrast to negotiations for what became the ICC, NGOs and civil society played a relatively small role in the establishment of the two ad hoc tribunals and in promoting related proposals. They were discussed and, in the case of the ad hoc tribunals, established by states as part of a broader ambition to promote, at least officially, not only justice, but also peace, security and stability.

The Rome Statute was negotiated in a context in which law was seen as a viable, although still potentially controversial, international governance mechanism. A range of different states invested in criminal law as a governance tool, although they often had distinct conceptions of how it ought to function. For instance, African states saw negotiations regarding the ICC as an opportunity to develop legal frameworks that were less dominated by global north powers and perspectives (Brett and Gissel 2020; Gevers 2020; Gissel 2018; Nimigan 2021). In a context where the two ad hoc tribunals successfully worked to secure prosecutions of gross human rights violations, the creation of the ICC was widely supported by human rights NGOs. In contrast to the ad hoc tribunals that were negotiated among states without significant NGO involvement, civil society representatives were not only present but extremely active at the Rome conference (Glasius 2006), a development that reflected a larger turn from advocacy and criticism to the direct support of prosecutions (Engle 2014, 2016; Lohne 2019). Established human rights NGOs such as Amnesty International (AI), Human Rights Watch (HRW) and the International Federation for Human Rights (FIDH) began efforts to shape and promote legal mechanisms to address human rights violations. New NGOs were established that specialized in supporting prosecutions. The establishment of the ICC was also closely observed and studied by

academics who increasingly developed publication profiles that focused on international criminal law and legal mechanisms to protect human rights.

The convergence between state investments, civil society growth, and scholarly engagement shaped an emerging view that international criminal law mechanisms could be deployed to promote justice. Legal mobilizations to internationalize the enforcement of crimes transpired across these different domains and involved elite agents from each of them. The field of international criminal justice is the product of these mobilizations. Whereas the interwar period and the Nuremberg trials focused on aggressive war as the main scourge, attention in this field shifted toward the goal of securing justice for victims of mass atrocities. Written into the preamble of the Rome Statute and popularized by NGOs, attention focused on a mission to end impunity (Houge and Lohne 2017). Multiple international criminal justice institutions were created and highlighted by civil society, academia, and the mass media. Public narratives linked core international crimes to the worst atrocities imaginable and popularized the notion of making those responsible criminally liable through mechanisms that could, at least in theory, end impunity. Mobilizations around core crimes built on 'injustice frames' (Gamson et al. 1982; Gamson 2013) that identified specific human rights problems, their victims, and the need for justice. The implementation of international criminal courts was identified as the solution to the problem. In the case of atrocity crimes, the *diagnostic*, *prognostic,* and *motivational* aspects were interwoven to the point where the solution to the problem of impunity, i.e., international courts, was viewed as the means to achieve justice for victims of atrocities, often assuming that an internationalized form of criminal justice was, in fact, what victims wanted.

The concurrent investments of states (for instance, through the secondment of staff to the international criminal courts), NGOs, and academics helped to create and define the contours not only of the ICC, but the broad field of international criminal justice focused on atrocity crimes. The turn towards criminal law coexisted with a form of liberal internationalism (Buchan 2013; Joyce 2016) that relied heavily on law as a governance mechanism. This was epitomized by the creation and proliferation of new international courts and the increased activity of international and regional courts established after the end of World War II (Alter 2014; Alter et al. 2016). The increased focus on international law, international governance, and international criminal justice, in a period characterized by the introduction of novel legal mechanisms to address atrocity crimes, facilitated discussion of new ideas and legal interpretations. In this context, some activists argued for a broad interpretation of the ICC's subject-matter jurisdiction that would include violations of specific treaty crimes (Boister 1998).

As these crimes were left out of the Rome Statute, the International Criminal Court (ICC) was given the power, under specific circumstances and adhering to the principle of complementarity (Nouwen 2013), to investigate, prosecute and adjudicate core international crimes. However, the capacity of the court depends on the cooperation of states parties, specifically their police powers (Peskin 2008). This a weakness. Currently, the field of international criminal justice is dominated by legal frames and professionals that are unmoored from the operational aspects of national or transnational policing (Christensen and Levi 2017; Christensen 2021; Levi and Hagan 2006). Many of these legal professionals move across the various international criminal courts, and thus became repeat players who define and shape the dominant narrative, practices, and discourse of the field. These agents were also active in later mobilizations around international criminal courts, with some engaging in the push to internationalize the prosecution of corrupt practices.

The legal views, visions, and preferred technologies of international criminal justice are written into the profiles of agents who work in this field and mobilize around its core premise of ending impunity by legal means. In academia, a new scholarly subfield of international criminal law was developed that focuses specifically on atrocity crimes (Christensen 2016; Cryer et al. 2019; Mégret 2020). At the same time, the international criminal courts are staffed with practitioners, many of whom specialize in international criminal law and have devoted decades of work to these institutions (Christensen 2015).

NGOs mobilize to monitor the development of these institutions, often criticizing them, while at the same time supporting prosecutions by facilitating access to witnesses and evidence (Dixon and Tenove 2013; Merry 2006). In recent years, some NGOs even specialized in gathering evidence that could be used for prospective prosecutions (Baylis 2009; Burgis-Kasthala 2019). UN Member States continue to invest resources and energy in these institutions, despite significant pushback (Madsen et al. 2018) against the ICC, especially from African states (Clarke et al. 2016; Clarke 2019), and sanctions implemented by former US president Donald Trump against ICC staff (Kreß 2020). In July 2021, for instance, Sudan announced its intention to become a party to the Rome Statute, becoming the 124th member state (Al Jazeera 2021).

## 4. Mobilizations to Internationalize Anticorruption Enforcement

Unlike other unsuccessful attempts to internationalize enforcement, such as terrorism and drug trafficking, the push to internationalize anticorruption enforcement has a relatively short history. This push reproduces conceptual frames developed around prior mobilizations to internationalize the prosecution of core crimes. The narrative, discourse, and injustice frames of the push to internationalize anticorruption enforcement closely parallel the narrative, discourse and frames employed to mobilize the international community in support of the international prosecution of core crimes.

The most highly debated suggestion to internationalize the prosecution of corruption comes from US Senior Judge, Mark Wolf. Wolf's original proposal was published in 2014 as a Brookings Institute Policy Paper (Wolf 2014) and, in 2018, Wolf published the proposal in *Dædalus*, the journal of the American Academy of Arts and Sciences (Wolf 2018). Wolf proposed the establishment of an International Anti-Corruption Court (IACC) and created a new NGO to mobilize support for the proposal, mirroring the creation of specialized civil society organizations in the field of international criminal justice. Wolf underlined the detrimental effects of grand corruption and kleptocracy, a perspective that derived partly from his own experience in judging cases involving corruption (Wolf 2014, 2018). The IACC is inspired by the ICC, which is tasked with the prosecution and adjudication of atrocity crimes through the appointment of new international prosecutors and judges. In presenting his proposal for the IACC, Wolf proposes that the IACC would assume complementary jurisdiction over corruption involving leaders of states parties to the IACC and anyone appointed by those leaders. He proposes that the IACC would also assume jurisdiction over cases involving the leaders of non-states parties when those states are unable or unwilling to investigate and prosecute. The proposal envisages that the IACC statute would empower the UN Security Council to refer cases to the IACC. In principle, this could involve states that were not states parties to the IACC convention. This mirrors the legal framework that structures the jurisdiction of the ICC.

In contrast to Wolf's call for an IACC, Stuart Yeh's proposal for an Anticorruption Protocol to the United Nations Convention against Corruption (APUNCAC) does not envisage the expansion of the subject-matter jurisdiction of the ICC or the creation of a new international court (Yeh 2021b). Instead, APUNCAC would establish dedicated domestic anticorruption courts funded by the UN. Yeh notes that, in principle, the IACC could serve as a supreme court for cases of grand corruption, if (and when) domestic courts are unable or unwilling to prosecute grand corruption (Yeh 2021c). APUNCAC would establish specialized UN anticorruption inspectors, a new group of professional criminal investigators who would work closely with national anticorruption courts. The UN does not currently field a permanent body of anticorruption investigators. However, the proposal builds on the precedent set by the bilateral agreement establishing the Commission against Impunity in Guatemala (CICIG). CICIG established a body of UN-funded and UN-supported investigators tasked with the investigation of high-level corruption in Guatemala and government-linked criminal gangs. APUNCAC would establish a similar body of UN-funded and UN-supported investigators tasked with the investigation of corruption affecting states parties to APUNCAC. These investigators would refer cases to dedicated

domestic anticorruption courts created and funded by the UN. Under APUNCAC, these national anticorruption courts would be periodically evaluated by the UN Commission on Crime Prevention and Criminal Justice. Funding would be redirected in cases where performance was not adequate. In summary, APUNCAC would expand the mandate of the UN to include criminal inspectors, modelled on a previous bilateral UN agreement. APUNCAC would give the inspectors powers to investigate corruption and refer cases to dedicated domestic anticorruption courts, and would give the UN the power to evaluate the UN-funded domestic anticorruption courts, redirect funding based on those evaluations and ensure accountability. The provisions of the proposal also address campaign finance, conflicts of interest, money laundering, the use of offshore accounts to hide illicit proceeds, and extraterritorial jurisdiction. APUNCAC establishes procedures to debar and extradite recalcitrant individuals and impose civil and criminal penalties.

Yeh's proposal was discussed in multiple academic articles over the past decade, following a broad tradition where research is used to test and advance potential solutions to world problems. A professor at the University of Minnesota, Yeh published several articles analyzing the rationale for APUNCAC as well as specific APUNCAC provisions designed to fight corruption (Yeh 2011, 2012, 2014a, 2014b, 2021a). The practice of publishing analyses of specific issues that may then be critiqued by other scholars reflects a wider, historically embedded, tradition of legal research, where existing legal frameworks are evaluated, weaknesses are assessed, and potential solutions are prescribed (Bourdieu 2000; Fallon 2012; Kahn 1999; Tushnet 1981). As such, legal scholars are often active stakeholders in formulating reform proposals, including ideas to reform the international legal regime. This practice is also visible in research regarding atrocity crimes (Byrne 2020; Christensen 2016; Manley 2016; Stappert 2018). However, whereas proposals for what became the ICC were supported by scholars such as Cherif Bassiouni, who had a central role in the space of academic production regarding international criminal law, the academic space that developed around anticorruption since the 1990s was more skeptical towards new proposals to internationalize anticorruption enforcement.

Indicative of the role of legal research as both prescription and critique, scholars debated the potential of implementing the IACC and APUNCAC. Wolf's 2018 article is partly a response to this criticism, and explicitly engages with some of these critics (Wolf 2018) as does Yeh's 2021 article (Yeh 2021c). Critics of the IACC include academics that view the proposal as unrealistic and ridden with flaws that would render its effectiveness questionable. Critics include corruption experts (Stevenson 2014) and, more generally, scholars of international law (Alter and Sorensen 2014) and of international criminal law (Whiting 2018). Much of this debate took place on the blogosphere, specifically the Global Anticorruption Blog, where supporters of the IACC proposal also published their views (Goldstone and Rotberg 2018).

Goldstone and Rotberg's (2018) blogpost in support of the IACC explicitly notes that mobilization efforts led to the creation of a supportive international NGO: Integrity Initiatives International (III) founded by Judge Wolf. Rotberg is the founding director of the Intrastate Conflict Program at Harvard University's Kennedy School of Government. Several other board members have close connections to Judge Wolf. Outside of being a board member of III, Goldstone has close links to international criminal law enforcement, having served as Chief Prosecutor for the International Criminal Tribunals for the former Yugoslavia (ICTY) and Rwanda (ICTR). In particular Goldstone's profile is often highlighted in III communications. His prominent role as an international jurist and ICTY and ICTR prosecutor lends weight to the push to internationalize the investigation and prosecution of corruption. III highlights this link and draws upon the connection of corruption to atrocity crimes to frames corruption as a social evil requiring urgent international action in support of the IACC. In general, III board members are well connected to legal practice, academia, industry and politics. In addition to board members with strong credentials from legal practice and academia (especially Harvard), III has close connections to the government and private industry. For instance, Camilo Enciso, who holds degrees from Harvard and

Colombia, is the founder and head of the International Institute for Anticorruption Studies in Bogota and has served as Secretary for Transparency in Columbia. Colombia has so far been the only country to voice support for the IACC.

In addition to III, and at times supporting its mission, other NGOs have focused on expanding the jurisdiction of the ICC or creating a new international criminal court to deal with corruption. For instance, the Global Organization of Parliamentarians against Corruption (GOPAC) organizes parliamentarians from the global south in the fight against corruption. GOPAC discussed the possible use of the ICC to prosecute corruption and voiced support for establishing the IACC to prosecute grand corruption (GOPAC 2013). Another NGO, the Public International Law and Policy Group (PILPG), also active in international criminal justice, organized a 2019 side-event at the Assembly of State Parties of the ICC where the expansion of the mandate of the court was discussed by practitioners and academics. Speakers included Oliver Windridge, representing The Sentry, an international NGO built around ideas that link corruption and atrocity crimes. The mission of The Sentry is to trace illicit funds of war criminals and transnational profiteers of war. The Sentry employs both investigative and policy personnel and has discussed the role of the ICC in tracing illicit funds of suspected war criminals (Dranginis 2019).

In addition to some NGO backing, limited political support has emerged for the IACC. In 2020, Colombia pledged to formally bring the idea of an IACC to the UN General Assembly 2021 Special Session focused on anti-corruption (Yeh 2021c), also referred to as UNGASS 2021. Colombia was later joined by Peru, which supported the move to establish an IACC. The proposal to establish such a court was also discussed at a UNODC Global Expert Group Meeting in Oslo, Norway, in 2019 (Trujillo 2019). The text adopted at the Oslo meeting did not offer support for the IACC specifically but did mention it alongside a range of options that could be explored (UNODC 2019). The UN General Assembly Special Session focused on corruption followed a similar sequence of events. New multilateral initiatives that could result in the creation of new institutions or legal frameworks played a minor role and were primarily discussed in the context of asset recovery (UNGA 2021). Colombia's own contribution at the General Assembly did not mention the IACC specifically, but underlined the need to strengthen the current system, including the possible creation of "new international institutions aimed at the pursuit and prosecution of kleptocrats and others responsible for acts of corruption internationally" (Colombia 2021). Other states remained reticent regarding the internationalization of anticorruption enforcement. However, the UNGASS 2021 political declaration did ask states parties to UNCAC to identify "any gaps and corruption challenges, and to consider any recommendations by states parties to address the gaps and challenges identified in such a way as to improve the Convention and the implementation thereof as may be necessary" (UNGA 2021).

However, whereas states generally did not voice support for internationalizing anti-corruption enforcement, NGO support for the IACC increased. Other than the contribution of III itself (III 2021), a range of other civil society organizations stated varying degrees of support for the idea of an international anti-corruption court or similar solutions (for instance, extending the mandate of the ICC). These organizations included TI (TI 2021), The Daphne Caruana Galizia Foundation (The Daphne Caruana Galizia Foundation 2021); the UNCAC Coalition, a coalition of 350 civil society groups that promote and monitor the ratification and implementation of UNCAC (UNCAC Coalition 2021); L'Observatoire de Lutte contre la Corruption et les Malversations Economiques (OLUCOME 2021); and the International Bar Association, although this submission mainly focused on asset recovery (IBA 2021). Several proposals urged UN Member States to consider the available options for international enforcement mechanisms, without naming specific options. In this context, the APUNCAC proposal was also represented at the conference and described the idea of employing an international body of UN inspectors to investigate charges of corruption and refer cases to dedicated domestic anticorruption courts (Yeh 2021a). It was the IACC,

however, that generated the greatest attention among NGOs. In this context, the role of civil society NGOs was central to the push to internationalize enforcement.

As part of this push, the use of injustice frames, similar to those employed to mobilize action regarding core crimes, are often employed by civil society NGOs to mobilize action toward international mechanisms to investigate and prosecute corruption. In general, two distinct but closely linked frames are employed. These frames link diagnostic and prognostic dimensions. The first relates to the conceptualization of corruption as a crime of international concern. The second draws upon established legal frameworks to underline the feasibility of the proposed solutions. The first frame aims to firmly establish corruption as a threat to crucial societal goods that are of international, perhaps even global, concern. The frames used to promote the internationalization of anticorruption enforcement highlight the role of corruption in exacerbating large-scale social, political, economic, gendered, and racialized imbalances within and between countries. Grand corruption is described as especially destructive and destabilizing. Internationalized enforcement is linked to the concept of ending impunity (Wolf 2014; Yeh 2021c), thus repurposing the rallying call around atrocity crimes to mobilize action against corruption. In this way, internationalized enforcement is justified by reference to the endemic nature of corruption, its detrimental effects, and its frequent association with other forms of illicit activity.

However, unlike mobilization processes that developed such frames around the internationalized enforcement of atrocity crimes, the mobilizations to internationalize anticorruption prosecutions are relatively narrow. They are not deeply rooted in influential academic circles, have little political support and are not strongly embedded in anticorruption-related professional structures or across integrity warriors involved in the monitoring and implementation of UNCAC. In addition, the activists who do seek to internationalize anticorruption enforcement deploy injustice frames borrowed from atrocity justice mobilizations but do so in a political context where the legal consciousness and ideology supporting international intervention is relatively weak. Mobilizations to internationalize the investigation and prosecution of corruption conflict with recurring notions that, within each state, domestic police and criminal justice systems are the preferred, legitimate institutions to enforce criminal justice. These notions are currently guarded more jealously than was the case in the 1990s where the international political community created multiple international criminal courts to target the investigation and prosecution of atrocity crimes. In addition, for many states, the internationalization of anticorruption enforcement may be more controversial than the creation of international criminal courts.

## 5. Discussion

The push to internationalize anticorruption enforcement often copied the legal formats and injustice frames employed by activists to mobilize international action regarding core crimes. While some of these frames were employed for nearly a century, they gained political support through concurrent mobilizations by activists, politicians and academics to build consensus around the ICC and similar institutions. The push to internationalize enforcement broadened to include anticorruption professionals linked to activist NGOs. Anticorruption activists employed scholarly publications and formed NGOs to mobilize action around corruption as an international crime. At the same time, established NGOs, both focused on anticorruption and to a certain extent on core international crimes, recently began to support the idea of an IACC or similar proposals. These mobilizations were framed and legitimized by reference to the detrimental effects of corruption and a motivation to end impunity. However, while these conceptual frames echo language that was employed regarding atrocity crimes, underlining the pernicious consequences of corruption, the patterns of mobilization to internationalize anticorruption enforcement are relatively narrow, both regarding their links to academia and embeddedness in professional groups working to investigate and prosecute these crimes.

These mobilizations have yet to activate state support—the ultimate goal of this type of push for action. Proposals for new international conventions can only advance

when potential states parties agree that new conventions are needed, lend their support for new conventions, participate in drafting such legal frameworks, and sign and ratify them. Presently, only a few states, primarily from the global south, have demonstrated interest in advancing international anticorruption enforcement. In the absence of state action, the proposals are currently struggling to gain traction. If history is a guide, these proposals may continue to serve as foci for activists seeking to mobilize UN Member States around international action to combat corruption. In the same way, activists sought to advance proposals for an international criminal court throughout the Cold War. However, anticorruption activists who seek to mobilize the international community face critics who argue that there is a lack of consensus for international law solutions among UN Member States. Critics note that corruption is especially difficult to address when illicit proceeds are moved to offshore havens beyond the reach of authorities pursuing those proceeds. Critics also note that international cooperation is often slow and difficult. They are not optimistic that internationalization, which depends on high levels of cooperation, is the answer. This perspective is reminiscent of the criticism and challenges faced by the international criminal courts in bringing international criminals to justice for core crimes. Despite this criticism, these institutions, including the ICC, have, in multiple cases, demonstrated some capacity to bring criminals to justice, albeit very slowly and in ways that have led to significant criticism. However, as pointed out by critics of the IACC (Whiting 2018), the push for the IACC (and related proposals) is developing in a context that is very different from that of the ICC. The international consensus that led to the Rome Statute and the patterns of mobilization on which these negotiations were built are missing.

In the current context, characterized by a lack of progress, one option for supporters of international anticorruption enforcement would be to abandon the effort to implement new international laws and institutions designed to address the problem of corruption. A second option would be to alter the direction of international anticorruption enforcement efforts and base it on ideas and frameworks that are different from, but possibly complementary to, the IACC. This second option is illustrated by APUNCAC. An ambitious initiative, APUNCAC is based on the premise that current mobilizations can push the international community towards creating new forms of internationalized enforcement. To produce conceptual legal frames that can support further mobilizations, APUNCAC differs from Judge Wolf's IACC in important ways. Most significantly, Yeh's proposal spells out a multitude of operational details—how cooperation with investigators would be secured, the consequences of noncooperation, and how extraterritorial jurisdiction would be secured. The rationale underlying this proposal is that domestic authorities are stymied when criminals use the international financial system to hide illicit funds in offshore havens. APUNCAC is based on the idea that addressing this type of corruption may require increased international cooperation. In this view, the need for increased international cooperation implies the need for a new international convention that formalizes and institutionalizes the type of cooperation that is provided, the form and manner of cooperation, and the consequences of failure to cooperate.

The first option (abandoning the effort to internationalize anticorruption enforcement) expresses a pessimistic view. Regarding this option, internationalization was tried and largely failed. This view suggests that further attempts to internationalize anticorruption enforcement are unlikely to gain the level of support that resulted in the international criminal tribunals for Rwanda and the former Yugoslavia and secured passage of the Rome Statute. In addition, this pessimistic view is also likely to underline what is perceived as the limited success of these institutions. The second option expresses a more optimistic view. In this view, despite skepticism in other circles, the internationalization of efforts to investigate and prosecute corruption can be redirected, with a focus on the operational details of assembling a body of UN inspectors, implementing dedicated anticorruption courts, creating incentives for private parties to pursue civil actions when those parties have knowledge of corruption, and implementing vigorous anti-money-laundering regulations. These competing views, pessimistic and optimistic, are rooted in disagreements among

legal scholars and anticorruption specialists regarding the most effective strategies for fighting corruption. Certain activists are hopeful that further internationalization will occur and are convinced that this is the most productive path towards the effective investigation and prosecution of corruption. Other anticorruption and international criminal justice specialists see internationalization as impotent regarding this crime and point to the pitfalls of this path. It remains to be seen whether the push to internationalize anticorruption enforcement will ultimately be successful, or if these efforts will fail. If mobilization is successful, and an international agreement is implemented, the ultimate question is whether the mechanisms implemented by the agreement are effective in fighting corruption.

The current article contributed an analysis comparing the mobilizations to internationalize the investigation and prosecution of core international crimes with those pushing for the internationalization of anticorruption enforcement. The article also compared the two contexts in which these mobilizations occurred, underlining their differences. The early efforts were supported by a fairly robust consensus that international action was needed. In contrast, current efforts lack this consensus and, perhaps linked to this contextual weakness, mobilizations aimed at internationalizing anticorruption enforcements efforts are narrow.

The implication of the analysis is that the context surrounding current efforts to internationalize the investigation and prosecution of (especially grand) corruption is not as supportive as the context that drove the creation of what became the field of international criminal justice. Since new international conventions can only advance when potential states parties lend their support to such legal frameworks, participate in their drafting, sign and ratify them, the context in which mobilizations occur matters. APUNCAC presumes that the present lack of consensus can be overcome. However, it is unclear whether the current context provides adequate support for the mobilizations that aim to advance APUNCAC, and, if not, what transformations must occur before this and similar proposals might gain traction in the international political community.

**Funding:** This research received funding from the European Research Council (ERC) under the European Union's Horizon 2020 research and innovation programme (JustSites, Starting Grant 802053).

**Institutional Review Board Statement:** The study was conducted in accordance with the relevant ethical guidelines, including the Danish Code of Conduct for Research Integrity, and has been approved by the Faculty of Humanities' Research Ethics Committee, University of Copenhagen (23 January 2019).

**Informed Consent Statement:** Not applicable.

**Data Availability Statement:** Not applicable.

**Conflicts of Interest:** The author declares no conflict of interest.

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
