# Peer review of "Legal Mobilization and the Internationalization of Anticorruption Enforcement"

_laws, 2020_

Round 1

Reviewer 1 Report

Paper has the potential to say something interesting and new, but there are several important shortcomings:

  • The aims of the paper should be clearly stated in the introduction, authors should also more deeply explain how their paper contributes to the current theory;
  • It would be recommended to emphasize the challenges and vulnerabilities related to the itinerary of internationalizing anti-corruption enforcement;
  • Given the fact that in the last decade has been a growing tendency to include anti-corruption provisions in new or revised national constitutions, the authors should discuss the subject from this point of view and to add some points related to the internationalizing anti-corruption enforcement and the status of permutations from the national level;
  • As the authors stated in the Abstract, “this article investigates characteristic mobilizations around these initiatives, the ways in which mobilizers frame their engagement, and the ideological context in which they operate”, it results that the paper needs to focus more on the context of legal mobilization and the internationalization of anticorruption enforcement, by including some case studies and critical analysis.
  • Methodology –authors should indicate the context of the qualitative sequential methodology applied;
  • Discussion-authors should discuss their results in comparison of relevant literature from web of science;
  • Conclusion-authors should make highlight their contribution to the theory and deliver some concrete recommendation for public policy (current recommendations are too general) and discuss new windows for possible research.

Author Response

First of all I would like to thank the reviewer for constrictively engaging with the article. In revising the article, I have focused in particular on making its contribution to the literature clear, as well as on making sure that this clarity is visible throughout the text.

In the revised article, the introduction has been considerably strengthened, something that also allows the text to make the challenges and vulnerabilities of the mobilizations around the anticorruption enforcement more clear. The revised conclusion now has new paragraphs that outline precisely some of these challenges and vulnerabilities as linked to the larger political context in which mobilizations take place.

As visible in the introduction and conclusion, the revised article highlights more clearly the importance of the historical and political context in formatting the mobilizations around internationalizing anticorruption enforcement as well as the previous push to internationalize the enforcement of atrocity crimes. These two efforts of mobilization around specific crimes serve as the case studies of the article. Since the focus in specifically on the agents pushing to internationalize enforcement, the injustice frames they develop to affect the political debate and the context in which this takes place, I have not found space to discuss also the increase of anticorruption norms and provisions in national constitutions. Although I agree this is an important recent development, I think a longer discussion about its links to efforts to internationalize enforcement is better placed in another article.

The revised article has also strengthened its presentation of the methodological framework used. Specifically, the article now outlines how it uses a historical sociological approach to analyzing the two case studies. As previously stated, the revised conclusion makes a more clear contribution to the literature, something that is made possible by implementing the changes outlined above in the article.

Reviewer 2 Report

Please consider changing the wording. For example, you wrote in the abstract:  "the ways in which". Such a phrase may be wordy. Try to change it using the phrase "how".  The phrase "anticorruption". Please consider "anti-corruption." 

Abstract: Please indicate your contribution to science more precisely. What does the study contribute to the current state of knowledge on the fight against corruption?

I suggest enhancing the literature review in the case of corruption, with references from researchers' broadest possible cultural circle. I mean many researchers from Africa, Asia and Europe, and here, for example, works highlighting problems in the fight against corruption and public ethnocentrism catalyses corruption and has the same effects as corruption. The phenomenon of corruption needs to be seen more broadly through the prism of money laundering and the preparation of various institutions to fight corruption. I just reviewed couple weeks ago the literature about corruption considering authors from countries experienced by corruption and I suggest analyse for example the following works:  https://doi.org/10.9770/jesi.2020.8.1(36)

https://www.ersj.eu/journal/1821

Combating corruption and other organizational pathologies Peter Lang GmbH Germany

https://doi.org/10.3390/su12010244

Competitive role of supreme audit institutions in building trustworthiness for customers

https://doi.org/10.14254/2071-8330.2020/13-2/13

These are just some examples of works on corruption. Such phenomenon, like corruption, cannot be eliminated without creating a system of cooperating entities: public, private, third sector, and international organizations. This phenomenon cannot be limited by dealing only with a fragment of the phenomenon. And yet, apart from bribery, it is also nepotism, cronyism, and so on. In addition to the literature mentioned above, one may also find different articles, books. Here is an opportunity in this article to present a comprehensive analysis of various concepts. I propose not limiting your literature study to only known surnames but also looking for authors from Asia, for example, Turkey. From Europe, for example from Italy, etc. The literature of authors from Central and Eastern Europe is fascinating. It is because they have experienced corruption three times. First, during the communist era, when bribery was a form of payment for unavailable consumer goods. Secondly, in the period of political and economic transformation, when no one in the world, apart from these countries, moved from socialism to capitalism. Finally, corruption is typical of an economy based on free-market competition. These experiences must be collected and presented appropriately.

Please consider formulating research problems. Please take into account newer literature and scientific achievements in the fight against corruption.

Please present the following in an orderly manner in the discussion:
1) contribution to the discipline and what discipline
2) contribution to the practice what needs to be done (please specify)
3) what are the limitations of research?
4) what are the directions of future research, and why?

Author Response

First of all, I thank the reviwever for engaging with the article. Alongside implemeting many of the comments of reviewer 1 and the editor of LAWS, the revised article takes on board some of the suggestions of reviewer 2. I have not, however, implemented all of the suggestions of reviewer. I will address the comments of the reviewer below, explaining also where I have not followed the advise of the review.

I agree that a simpler wording than “the ways in which” is a better fit for the article, especially since the two cases in which this phrasing was used were both in the abtract. With regard to anticorruption, however, I adhere to the rules of the journal.

The reviewer makes a good point that researchers from the global south have important voices with regard to anticorruption. I generally agree that researchers in the global north have to be aware to cite scholars representing countries from the global south. So far, however, the literature on mobilizations to internationalize enforcement of specific crimes has been published by the authors cited in the article, many of whom work in institutions in the global north.

Outside of scholarship, the article does in fact focus on the role of global south agents in mobilization efforts around the enforcement of anticorrution and atrocity crimes. The role of the Soviet Union, Caribbean and African nations is discussed with regard to the push to internationalize enforcement of atrocity crimes, and the role of specific countries in Latin America, as well as NGOs active in the global south, is discussed with regard to the ongoing mobilization around the internationalization of anticorruption enforcement. I hope that the article can  open avenues for future research of global south researchers, or research collaborations that cut across global divides.

As part of the argument to bring in more global south literature, the reviewer recommends specific publications. Unfortunately, the recommendations all point to the same scholar. I have not cited any of the listed works.

The reviewer makes that point that the phenomenon of corruption cannot be dealt with by analyzing only “a fragment of the phenomenon” and recommends that the article presents “a comphenensive analysis of various concepts”. Concepts mentioned by the author include bribery, nepotism and cronyism. While I agree that corruption is a complex phenomenon, I do agree that the article ought to focus on conceptual distinctions more broadly or on the experiences of authors in former communist countries (as implied by the statement of the reviewer that “These experiences must be collected and presented appropriately”). This would simply be a very different article and not the one I was asked to contribute to this special issue.

The reviewer concludes by asking for four very broad revisions. As highligted in the response to the editor and reviewer 1, the revised article has considerably strengthened its introduction and conclusion, as well as crucial elements of the text, for instance its presentation of used methods. I believe the implemented changes also adress the final comments of reviewer 2.

Round 2

Reviewer 2 Report

Please expand the literature about combating organizational pathologies, including corruption, fraud, money laundering. What others wrote about it? Make sure, that the part of article with liturature review is also valuable. 

Author Response

Thanks for your comments.  The literature on organizational pathology and corruption leads me directly to the same author that the reviewer referred me to last round. I will not cite specific authors in this way.